# *Lactobacillus* Bacteremia and Probiotics: A Review

**DOI:** 10.3390/microorganisms11040896

**Published:** 2023-03-30

**Authors:** Ravina Kullar, Ellie J. C. Goldstein, Stuart Johnson, Lynne V. McFarland

**Affiliations:** 1Expert Stewardship Inc., Newport Beach, CA 92663, USA; 2R.M. Alden Research Laboratory, Santa Monica, CA 90404, USA; ejcgmd@aol.com; 3Edward Hines Jr. VA Hospital, Hines, IL 60141, USA; 4McFarland Consulting, Seattle, WA 98115, USA

**Keywords:** bacteremia, sepsis, probiotics, *Lactobacillus*

## Abstract

Lactobacilli are widely found in nature, are commensal microbes in humans, and are commonly used as probiotics. Concerns about probiotic safety have arisen due to reports of bacteremia and other *Lactobacillus*-associated infections. We reviewed the literature for articles on the pathogenicity of *Lactobacillus* spp. bacteremia and reports of probiotics in these patients. Our aim is to review these articles and update the present knowledge on the epidemiology of *Lactobacillus* spp. bacteremia and determine the role of probiotics in *Lactobacillus* bacteremia. *Lactobacillus* bacteremia is infrequent but has a higher risk of mortality and risk factors, including severe underlying diseases, immune system suppression, admission to intensive care units, and use of central venous catheters. A variety of *Lactobacillus* species may cause bacteremia and may or may not be associated with probiotic exposure. To determine if oral probiotics are the source of these infections, the blood isolates and the oral probiotic strain(s) must be compared by sensitive identification methods. The prevalence of *Lactobacillus* bacteremia is infrequent but is more common in patients taking probiotics compared to those not taking probiotics. Three probiotics (*Lacticaseibacillus rhamnosus* GG, *Lactiplantibacillus plantarum*, and *Lacticaseibacillus paracasei*) were directly linked with blood isolates from bacteremia patients using molecular identification assays.

## 1. Introduction

*Lactobacillus* species are Gram-positive, non-spore-forming, rod-shaped facultative bacteria that ferment carbohydrates into lactic acid as a major endpoint [1]. These bacteria are taxonomically complex and often require molecular identification for speciation. They are widely distributed in nature and have multiple commercial uses, especially in the fermentation of cheese and other dairy products [2,3]. Many lactobacilli are commensal flora found within the human gastrointestinal tract and the female genitourinary tract. They are beneficial in helping protect from chronic diseases such as inflammatory bowel disease and are an important member of the gut and vaginal microbiome. Further, since most *Lactobacillus* species are probiotic microorganisms, they produce enzymes that display antibiotic, anticancer, and immunosuppressant properties [4]. However, in some cases, lactobacilli can act as opportunistic pathogens causing *Lactobacillus* bacteremia (LB), endocarditis, liver and dental abscesses, and prosthetic knee infections [2,5,6,7,8]. The source of these infections has sometimes been attributed to the use of *Lactobacillus*-containing probiotics [9,10,11]. Probiotics are commonly used for the prevention of *Clostridioides* (*Clostridium*) *difficile* infections or the prevention of antibiotic-associated diarrhea, among other diseases [1,12,13]. Although probiotics are generally recognized as safe, and most have excellent safety profiles, concerns about their safety have arisen due to reported cases of LB [14,15,16].

In order for probiotics to be appropriately attributed as an index source for *Lactobacillus* spp. infections, it is necessary to utilize current genomic identification methods and match the blood isolates with the potential source probiotic, using the current nomenclature for both isolates. Recently the identification of the genus *Lactobacillus* has been divided into 23 genera, >250 species, and 57 subspecies [17]. As a consequence, updated nomenclature for *Lactobacillus* spp. needs to be linked with older names in published articles.

In this paper, we will review the reported cases of LB, their epidemiology, and examine the role of *Lactobacillus*-containing probiotics in LB.

## 2. Materials and Methods

We reviewed the literature from 1980 to 2023 for articles on the pathogenicity of *Lactobacillus* spp. bacteremia and reports of probiotics in these patients. Search terms included *Lactobacillus*, bacteremia, probiotics, risk factors, treatment, antibiotic susceptibilities, safety, epidemiology, sepsis, endocarditis, and infection. Inclusion criteria included case reports, case series, observational studies, and reviews, while exclusion criteria included studies on non-*Lactobacilli* bacteremias, fungemias, or other types of diseases. We also queried references cited from reviews and published papers.

## 3. Results

### 3.1. Prevalence of LB

The prevalence of LB is generally low, comprising 0.1–0.2% of all isolates identified in blood cultures from hospitalized patients and 0.5% of blood isolates from immunocompromised patients [2,10,18,19,20]. In one study of a cohort of hematopoietic cell transplant recipients at a hospital in Seattle, Washington, between 2002 and 2011, the incidence of LB was 1.6 cases/100,000 patient days [10]. Salminen et al. investigated whether there was an increase in bacteremia caused by *Lactobacillus* spp. in Finland after *Lactocaseobacillus (Lcb.) rhamnosus* GG was introduced in dairy products in 1990 and reached common use by 1999 [20]. However, even when the use of this probiotic increased, no increase in the incidence of LB was found, with an average incidence of 0.29 cases/100,000 inhabitants/year between 1995 and 2000. LB may be underdiagnosed, as lactobacilli are difficult to culture and identify, and, in many cases, have been deemed as contaminants in blood samples [2,21].

### 3.2. Etiologies of LB

The most common species isolated from LB blood samples is *Lcb. rhamnosus* GG (ATCC53103) [6,7,15,22,23,24]. Salminen et al. reviewed 89 patients with LB, and polymicrobial bacteremia was found in 39% of patients, and ≥2 additional bacteria other than lactobacilli were isolated from 12% of patients [9]. Cannon et al. found that in 129 cases of LB, most were due to *Lcb. casei* (36%) or *Lcb. rhamnosus* (23%), but did not report the strains involved [25]. Other *Lactobacillus* species have been less frequently reported: *Lcb. paracasei* [7], *Lactobacillus* (*L.*) *acidophilus* [7,15], *L. jensenii* [7,23,26], *Lactiplantibacillus* (*Lpb.*) *plantarum* [7,27], and *Limosilactobacillus* (*Lsb.*) *fermentum* [7,23]. Husni et al. reviewed 45 cases of LB and found among the 12 isolates that were identified by species, *Lcb. casei* was the most common *Lactobacillus* spp. identified (10/12) [6]. In one trial in immunocompromised HIV patients, *Lcb. rhamnosus* GG was given to 20 patients, yet none developed LB [28].

### 3.3. Risk Factors for LB

LB is usually associated with patients who have comorbidities or suppressed immune systems [6,7,9,28,29,30]. LB has been reported in patients with cancer [10,31,32,33], ischemic or ulcerative colitis [34,35], that are immunocompromised (HIV, chronic steroids, chemotherapy, or transplant patients) [36,37,38], or in preterm neonates with underlying conditions (short bowel syndrome, growth restriction) [39]. Table 1 displays the risk factors associated with LB.

Salminen et al. reviewed the risk factors and outcomes for LB using multivariate analysis and found severe underlying diseases, which were primarily malignancies or serious gastrointestinal disorders, and were a significant predictor for mortality (odds ratio [OR] = 15.8). In contrast, treatment with antibiotics effective in vitro was associated with lower mortality (OR = 0.22) [9]. In 82% of the cases, the patients had severe or fatal comorbidities. Husni et al. reviewed the cases of 45 patients with LB occurring for over 15 years and found 22 (49%) were in the intensive care unit (ICU) at the time of onset of LB [6]. Many patients (60%) had polymicrobial infections, with concomitant bloodstream isolates including streptococci, *Candida* species, and enteric Gram-negative bacilli. Patients were typically elderly (mean age, 61 years) and had significant underlying comorbidities, including cancer (40%), recent surgery (38%), and diabetes mellitus (27%).

A retrospective study of 38 patients presenting with LB between 2005 and 2014 by Franko et al. found 31 (82%) LB-positive patients had at least one risk factor, most commonly cancer (40%), immunosuppression (37%), use of central venous devices (29%); 39% of patients presented with ≥2 risk factors [30]. Case reports have similar findings to these studies, with LB as a complication of serious underlying conditions, including liver transplantation, AIDS, metastatic choriocarcinoma, and prolonged marrow aplasia [31,36,37,40].

### 3.4. Consequences of LB

Mortality in patients with LB has been estimated at 30%, but typically, the direct causes of death are attributed to underlying conditions or accompanying infections [25]. Salminen et al. found that in patients with LB, the mortality was 26% at one month and 48% at one year [9]. Husni et al. reviewed cases of 45 patients over 15 years with clinically significant LB [6]. Twenty-two patients (44%) died during their hospitalization; however, only one death was directly attributable to LB. At the year one follow-up, 69% of patients had died. In all cases, the cause of death was due to underlying conditions rather than the LB itself. Franko et al. found a very high case fatality in 38 LB patients by day 28 (23.7%) and at one year (45%) [30]. LBs were divided into two clinical–biological presentations: secondary bacteremia with a known portal of entry (n = 30) and isolated bacteremia (n = 8). The one-year case fatality was higher in the secondary bacteremia group compared to the isolated bacteremia group (55.2% vs. 12.5%; *p* = 0.021).

Few studies have evaluated the impact on the length of hospital stays due to LB. Salminen et al. reviewed the risk factors and outcomes for patients with LB [9] and noted that there were more days of hospitalization after the onset of LB compared to prior to the onset of bacteremia. A case-control study of LB found that patients developing LB extended their hospital stay by an average of 5 days compared to patients with no LB [41]. More studies evaluating the impact of LB on the length of hospital stay and healthcare-associated costs are warranted.

### 3.5. Treatment and Susceptibility

The treatment of LB should be guided by the clinical presentation and susceptibility testing results due to the variable antibiotic susceptibility pattern associated with *Lactobacillus* spp. Goldstein et al. reviewed the susceptibility of commonly isolated human lactobacilli by individual species [1]. They noted that it is impossible to predict or make specific recommendations about individual species or strains. In general, susceptibility studies of lactobacilli have used various and diverse sources (human vs. commercial vs. animal), methodologies (e.g., ETest vs. broth microdilution or agar dilution), culture media, atmospheres of incubation (e.g., augmented CO_2_ atmosphere vs. anaerobic atmosphere) and inocula, as well as different breakpoints to determine susceptibility and resistance. For example, some *Lactobacillus* spp. are intrinsically resistant to vancomycin and aminoglycosides, while other glycopeptides will have varying activity.

Typically, when lactobacilli grow in culture from patient samples, they are classified as contaminants and antibiotics are not administered to the patient. It is important, therefore, to pay careful attention to a patient’s signs and symptoms, with a thorough laboratory workup performed to confirm clinical infection. The most commonly used antibiotics include penicillins (penicillin and ampicillin), with or without aminoglycosides. In a retrospective study of 241 cases of *Lactobacillus* spp. infections, the most commonly used regimens included penicillin monotherapy (n = 35), penicillin treatment with aminoglycosides (n = 20), and cephalosporins in monotherapy (n = 16) [25]. Clindamycin (90.0%) and erythromycin (94.3%) had the highest susceptibility, and penicillin had 63.6%. Vancomycin had the greatest resistance rates (sensitive only in 22.5% of cases). Clinical experience and clinical trials on the optimal antibiotic treatments are lacking due to the complexity of *Lactobacillus* species identification and susceptibility testing, plus the infrequency of infections caused by lactobacilli.

Salminen et al. conducted a study comparing the antibiotic treatment and susceptibilities in 85 patients between 1984 and 2000 of LB, including 46 *Lcb. rhamnosus* strains and 39 other *Lactobacillus* spp., and found 83% of the patients received combination therapy. However, for 54% of them, therapy included only one microbiologically active agent, according to the results of the susceptibility tests [23]. All isolates demonstrated low minimum inhibitory concentrations (MICs) against imipenem, piperacillin–tazobactam, erythromycin, and clindamycin. The ranges for the MICs of cephalosporins, which are frequently used to treat bacteremia, were wide, and very high MICs were demonstrated for many isolates. In general, second-generation cephalosporin and cefuroxime showed greater activity against lactobacilli, compared with third-generation cephalosporin and ceftriaxone, which is in accordance with earlier observations.

### 3.6. Role of Probiotics in LB

Case reports of LB have hypothesized that, in some cases, the source of the lactobacilli was the oral probiotic the patient was taking. However, this correlation was based on medical history alone or microbiologic cultures or assays that were unable to distinguish different *Lactobacillus* strains, or the strains themselves were not reported; thus, the hypothesis could not be supported [41,42,43]. In one case of preterm neonatal bacteremia, a probiotic containing *Lsb. reuteri* (strain not reported) had been given, but the blood isolate was only tested using biochemical methods only and, thus, could not identify the specific strain of *Lsb. reuteri* involved [44].

#### 3.6.1. LB with No Prior Probiotic Use

Our review also revealed 49 case reports of LB without any history of probiotic use (Table 1) [24,26,27,33,35,45,46,47,48,49,50,51].

One study reviewed cases reported over 5.5 years at a hospital in Boston, Massachusetts, and found of 21,652 patients admitted to their ICU who were not taking any probiotics, only two (0.009%) cases of LB were detected, which were caused by either *L. acidophilus* or *Lpb. plantarum*, but the specific strain was not reported [47]. The LB cases had either prior surgery or were immunocompromised. The incidence of LB in non-ICU patients not prescribed probiotics was even lower (10/93,000; 0.0001%) and was caused by a variety of lactobacilli (*Lcb. rhamnosus*, *Lpb. plantarum*, *Lcb. Casei*, or *L. gasseri*), but once again, the strains were not reported. From a literature review of 241 cases of lactobacilli infections (129 cases of LB, 73 cases lactobacilli endocarditis, and 39 cases of localized lactobacilli infections), none were associated with probiotic exposure [25].

#### 3.6.2. Mis-Matching Probiotic and LB Isolate Cases

As LB can occur in patients with no probiotic exposure, it is important that when there is a probiotic exposure, that the isolate from the blood and the strain of the oral probiotic be accurately compared using sensitive molecular techniques, such as pulsed-field gel electrophoresis (PFGE), 16s rRNA gene sequencing, DNA fingerprint analysis (rep-PCR), and whole genome methods. These methods have variable discriminatory power, but all have been used to compare blood and probiotic *Lactobacillus* spp. isolates [23,38,47,52]. In order to determine the role of probiotics in LB, both the blood isolates and the probiotic strain(s) must be thoroughly identified and compared, but this has not always been carried out. Several studies have reported a risk of probiotic use and invasive infections with common probiotic organisms. In a retrospective, matched case-control study, 28 adult inpatients with invasive infections (including positive cultures in blood, spinal fluid, ascitic fluid, or pleural fluid) were matched with 84 controls with no invasive infections [41]. The invasive infections were primarily due to *Lactobacillus* spp. or *Bifidobacterium* spp., but the specific strains were not reported. They reported probiotic use was more common (7/28, 25%) in cases with invasive infections compared to controls (5/84, 6%). However, the type of probiotic was only identified as either “Culturelle, Kefir, or probiotic yogurt”, and specific probiotic strains were not reported. In addition, cases of LB were not analyzed separately, thus, the conclusions that probiotics may be associated with LB cannot be drawn.

When molecular identification techniques were used, three cases of LB were found where the *Lactobacillus* spp. strain isolated from blood samples did not match the identity of the oral probiotic being taken by the patient (Table 2) [53,54,55]. In one case report, an 81-year-old male with comorbidities presented with abdominal pain and distension and was admitted [54]. On admission, he received antibiotics and was started on an oral *L. acidophilus* probiotic. On Day 9, he developed sepsis and cholecystitis. *Lsb. fermentum* was isolated from the body sample, but this was different from the probiotic strain the patient was taking (*L. acidophilus*). Another report of a 69-year-old male admitted with heart failure who was treated with antibiotics and was also given an oral three-strain *Lactobacillus* spp. probiotic (*L. acidophilus* CL1285, *Lcb. casei* LBC80R, and *Lcb. rhamnosus* CLR2, Bio-K+**^®^**) to prevent *C. difficile* infection [53]. Four weeks later, he developed sepsis, but the blood isolate (*Lcb. casei*) had no genetic relation to the probiotic strain (*Lcb. casei* LBC80R).

Another example of mismatching isolates was reported by Brecht et al. when a premature neonate undergoing intestinal surgery and having a central catheter was given the probiotic “Infloran™” (a combination of *L. acidophilus* ATCC4356 and *Bifidobacterium bifidum* ATCC15675) from Day 18 to 59 of the admission [55]. On Day 63, the neonate developed sepsis and the blood isolate was confirmed by genetic typing as *Lcb. rhamnosus* (strain not reported). These cases show that despite patients taking oral probiotics, LB may not be due to the probiotic being taken by the patient.

#### 3.6.3. Matching Probiotic and Blood Isolate Strains

Reports were also found where the blood isolate from LB was identical to the strain in the probiotic taken by the patient. Our review of the literature (from 1980 to 2023) found 23 cases where the blood isolate from a bacteremic patient was identical to the probiotic strain the patient had been taking. These cases are summarized and ordered by publication date in Table 3 [11,38,42,47,48,56,57,58,59,60,61,62].

Kunz et al. reported two cases of LB in infants with short bowel syndrome [42]. One case was a 36-week gestation male infant with short bowel syndrome secondary to congenital intestinal atresia and volvulus. The infant was on total parenteral nutrition (TPN) and was given *Lcb. rhamnosus* GG starting on day 95 of life. After 23 days of probiotic administration, the infant developed sepsis and the blood isolate was identified as a *Lactobacillus* species. Although fingerprint identification was not used, the sepsis resolved after the probiotic was discontinued. The second case was a 34-week gestation male with gastroschisis and where a jejuostomy was carried out shortly after birth. The infant was also dependent upon TPN and *Lcb. rhamnosus* GG was begun on the 17th day of life. On day 186 of life, the infant developed sepsis, and blood cultures were positive for *Lactobacillus* and were confirmed to be *Lcb. rhamnosus* GG by DNA fingerprinting.

De Grotte et al. also reported a case of LB in a premature neonate with necrotizing enterocolitis and short bowel syndrome who was dependent upon TPN [56]. After developing rotavirus diarrhea, he was given *Lcb. rhamnosus* GG starting at 6 months of age. At 11 months of life, he developed a fever and hypoxia, which was treated initially with vancomycin and ceftazidime. Blood cultures were positive for *Lactobacillus* spp. and *Candida albicans*. The identity of the blood isolate was confirmed to be *Lcb. rhamnosus* by 16-s ribosomal RNA analysis patterns. The patient was treated with amphotericin B, ampicillin, and gentamicin, and the central catheter was removed; the LB symptoms resolved.

Land et al. reported two cases of infants with central catheters who developed LB during their hospitalization [57]. The first case was a 6-week-old male who was admitted for cardiac surgery, had a central venous catheter placed, and was administered broad-spectrum antibiotics. Antibiotic-associated diarrhea developed on Day 57 and *Lcb. rhamnosus* GG was started 22 days afterward. After 20 days, sepsis developed and blood isolates grew out *Lcb. rhamnosus* GG, which was confirmed with DNA fingerprinting. The symptoms resolved after the probiotic was discontinued, the central catheter was removed, and broad-spectrum antibiotics were given. The second case was a 6-year-old girl with cerebral palsy and other comorbid conditions who was fed through a gastrojeunostomy tube and was admitted with a urinary tract infection, which was treated with ceftriaxone. Enteral feedings were discontinued, and a central catheter was placed. On day 18 of the hospitalization, enterococcal sepsis developed and the patient was treated with additional antibiotics. *Lcb. rhamnosus* GG probiotic was administered starting on day 25 of hospitalization to prevent antibiotic-associated diarrhea. The patient developed a new episode of sepsis 44 days later, and the blood isolate was confirmed by DNA fingerprint analysis as *Lcb. rhamnosus* GG. The symptoms resolved after ampicillin treatment and discontinuation of the probiotic. She was discharged with no symptoms 17 days later.

Vahabnezhad et al. reported one case of LB in a 17-year man with ulcerative colitis [58]. He was on systemic corticosteroids and infliximab, and developed sepsis after one week of *Lcb. rhamnosus* GG; his symptoms resolved after antibiotic treatment. The blood isolate was confirmed by 16s rRNA sequence analysis to be identical to the probiotic strain he was taking.

Sadowska-Krawczendo et al. reported one case of LB in a 6-day-old infant with intrauterine growth restriction [59]. The infant was treated empirically with antibiotics and *Lcb. rhamnosus* GG was given to prevent antibiotic-associated diarrhea. After six days, the neonate developed sepsis and the blood isolate was verified as *Lcb. rhamnosus* GG with 16s rRNA sequencing. The symptoms resolved after treatment with antibiotics, and the patient was discharged after 27 days.

Meini et al. reported a case of LB which developed in a 64-year-old female with ulcerative colitis [60]. She was admitted with severe diarrhea and also had coronary artery disease. During her admission, she developed a fever and was treated with multiple courses of antibiotics. Her blood culture grew methicillin-resistant *Staphylococcus aureus* (MRSA), which was treated with vancomycin. In addition, to restore her intestinal microflora, *Lcb. rhamnosus* GG was also given. Symptoms initially resolved, but 13 days later, the fever returned. Blood isolates grew out *Candida albicans* and *Lcb. rhamnosus* GG (confirmed by pulsed-field gel electrophoresis (PFGE). The patient was treated with amoxillicin–clavulanate and caspofungin and responded to this treatment.

Dani et al. reported two cases of LB in preterm neonates who were given *Lcb. rhamnosus* GG to prevent either antibiotic-associated diarrhea (AAD) or the development of necrotizing enterocolitis (NEC) [11]. Both neonates were hospitalized and received multiple courses of antibiotics and had central venous catheters. The first case was a 39-week gestation female who was on mechanical ventilation and had cardiac problems. She was given *Lcb. rhamnosus* GG starting on day 9 to prevent AAD, but developed sepsis on day 97. Her blood isolate was confirmed to be identical by PFGE to the probiotic she was taking. The second case was a 23 week gestation male who had patent ductus arteriosus and had a central catheter. *Lcb. rhamnosus* GG was started on the second day of life to prevent NEC. After 16 days, sepsis developed, and the blood isolate was confirmed by PFGE to be *Lcb. rhamnosus* GG. Both neonates were treated with antibiotics and survived.

Koyama et al. reported one case of septic shock associated with consumption of a yogurt containing *Lcb. rhamnosus* GG [61]. This occurred in a 54-year male with acute promyelocytic leukemia who had received an autologous stem cell transplant and was receiving high-dose chemotherapy. After developing severe diarrhea, he started a probiotic yogurt but developed septic shock one week later. The blood isolate was confirmed by PCR analysis as *Lcb. rhamnosus* GG.

Yelin et al. reported that 6 of 522 (1.1%) pediatric patients admitted to an ICU at a Boston Massachusetts hospital and receiving the probiotic *Lcb. rhamnosus* strain GG developed *Lactobacillus* bacteremia compared to only 2 of the 21,652 (0.009%) patients who did not receive *Lcb. rhamnosus* GG [47]. In addition, they identified blood isolate mutations that did not appear in the probiotic isolates, suggesting de novo evolution within the patient.

Cavicchiolo et al. described an interesting cluster of three cases of LB at a neonatal ICU [48]. The index case was a 25-week gestation female born with low birth weight (770 g) who started enteral feeding at 48 h and had a peripherally inserted central catheter (PICC) placed. On day 3 of life, she was also given a probiotic (*Lcb. rhamnosus* GG). After 15 days, she developed abdominal distension and sepsis. Enteral feeding, as well as the probiotic, were discontinued, and broad-spectrum antibiotics were begun. Her symptoms resolved after antibiotics were administered and she was discharged later. Her blood isolate was confirmed by PCR as *Lcb. rhamnosus* GG. Two other neonates in the same room as the index case also had a PICC placed and developed sepsis but had not been taking any probiotics. However, the blood isolates in both cases were confirmed by PCR to be identical to the probiotic (*Lcb. rhamnosus* GG) that their roommate was taking.

Chiang et al. reported one case of a preterm female who had low birthweight (749 g) and had respiratory distress syndrome [62]. She was given empirical antibiotics and had a PICC inserted. To prevent the development of NEC, *Lcb. rhamnosus* GG was begun on day 14. On day 28, she developed sepsis and NEC and was treated with vancomycin, cefotaxime, and metronidazole. The blood isolate and the isolate from the PICC catheter tip were confirmed by genomic sequencing assay to be to identical to the probiotic *Lcb. rhamnosus* GG strain. She recovered after being treated with ampicillin and pipercillin/taxobactam.

Gilliam et al. reported a cluster of six pediatric hematopoietic cell transplant recipients (aged 10 months to 16 years old) admitted to an ICU who developed LB and had been prescribed probiotics [38]. A high potency (1 × 10^11^ bacteria/day) probiotic containing seven types of bacteria (*Strept. thermophilus*, *Bifido. breve*, *Bifido. lactis*, *L. acidophius*, *Lpb. plantarum*, *Lcb. paracasei*, and *L. helveticus*) was given to 34 patients during their hospital stay to prevent any nosocomial infections. From January 2017 to December 2019, 6/34 (17.6%) children given the probiotic developed LB (4.7/1000 patient-days). After identifying the blood isolates using core-genome sequence typing, three patients had blood isolates matching at least one bacterial species found in the oral probiotic blend (*Lcb. paracasei* in one case and *Lpb. plantarum* in two cases). Unfortunately, neither the specific strains nor the identity of the proprietary probiotic blend were reported.

In our review, *Lcb. rhamnosus* GG was the most common etiologic strain found in these 23 cases. *Lcb. rhamnosus* GG (ATCC 53103) is more easily identified by its unique large colonies and the better availability of molecular methods. It is also widely available and used worldwide. In 1314 trials registered at ClinicalTrials.gov by 2019, *Lcb. rhamnosus* GG was the most common probiotic studied (146 trials) [63].

### 3.7. Safety of Probiotics

Previous reviews and clinical practice guidelines have urged caution for probiotic use due to safety concerns [7,16,64]. However, a causal link between the type of oral probiotic taken by the patient and the identity of the strain isolated in the patient with LB is infrequently documented. As shown above, the majority of the reported cases of LB did not have supporting evidence that the probiotic strain and the blood isolate were identical. Of the 75 total cases of LB from our review, only 23 cases were linked to an identical strain found in the probiotic taken. A review of 1569 studies reporting sepsis in preterm neonates found only 32 cases associated with either *Bifidobacterium* or *Lactobacillus* strains [39].

The safety of oral probiotics can also be determined with prolonged surveillance studies at hospitals where probiotics are used. Two long-term surveillance studies followed patients given antibiotics and a three-strain probiotic blend (*L. acidophilus* CL1285, *Lcb. casei* LBC80R, *Lcb. rhamnosus* CLR2, BioK+^®^) were followed for any complications of probiotic use [65,66]. No cases of LB were found in 44,850 adult patients over a 10-year period (2005–2014) surveillance period [65], nor in 4543 patients (from 2016–2019) [66]. Given the worldwide market for probiotics (in 2020, USD 34.1 billion and expectations that it will reach USD 72.9 billion by 2030), it is important to document the safety of probiotic use [67]. More long-term surveillance studies documenting probiotic use and LB are warranted.

Even though the risk of LB in patients taking probiotics is low, vigilance should be undertaken when probiotics are used in vulnerable populations (e.g., immunocompromised, premature neonates, use of central catheters). Additionally, the opening of capsules and reconstituting the powder to liquid for tube feedings increases the risk of LB not only for a patient, but also for their roommates, especially in the presence of central venous catheters [48,68]

## 4. Discussion

Our review found that Lactobacilli strains have been documented to cause bacteremia and sepsis in patients who are immunocompromised or hospitalized with severe disease. Antibiotic exposure during the hospitalization was common, which may have impacted the intestinal microflora and resulted in disruption of the intestinal barrier, allowing passage of intestinal bacteria to translocate out of the intestines into the blood. However, LB may occur with or without probiotic exposure. In the 23 cases of LB that were positively identified with the strains found in the probiotic the patients were taking, common predisposing factors seem to be exposure to broad-spectrum antibiotics, placement of central venous catheters, and severe morbidities. Preterm neonates with central catheters seem to be at higher risk, but cases of adults developing LB due to probiotics were also found. We recommend taking a precautionary approach to using probiotics in the ICU in patients with central venous catheters. The risk may be because powder more easily distributes to contaminate central venous catheters. The risk of contamination is likely both from the air and contaminated hands, which can cause a translocation to a central line catheter where the organisms have direct entry into the blood.

The possible mechanisms of LB should be addressed. As a risk factor for LB is the presence of a central venous catheter, direct contamination of the central line with lactobacillus or with stools containing the strain could lead to LB. Further, most patients that have LB have severe underlying diseases that predispose them to bacteremia complications, including LB. It has been noted that adhesion to intestinal surfaces is one of the main selection criteria for probiotic bacteria, as well as a lack of platelet aggregation, as this has been shown to be harmful to the host and potentially leads to endocarditis. Adhesion allows preservation in the gut and increases the opportunity for probiotic species to influence the host’s microbial balance and gastrointestinal immune system [69,70,71].

Recent guides to improve reporting of probiotic studies stress the importance of reporting the complete identification of probiotic strains [72,73]. Molecular identification of the *Lactobacillus* spp. in titles of publications and when analyzing the association with a possible causative infection is paramount for scientific validity. Broadly inclusive titles should disseminate misinformation to scientists, clinicians, and the general public. Others have reported a lack of efficacy of “probiotics” when they studied only a specific strain that was not included in the title, creating misperceptions in physicians who only read the titles or “news summaries” [74]. Recently, several authors again raised concerns about the safety of probiotics, thus, it is important that LB be well documented [7,16].

## 5. Conclusions

The safety and potential role of probiotics in human disease continues to be debated. We found molecular identification of both the blood isolate and the probiotic taken by the patient with LB is paramount to determining the infection’s source. Even though *Lactobacillus* strains used in common probiotics have been isolated from LB patients, we found many cases where the patient had no history of probiotic use or the strains were different. Many reports attempting to link the probiotic strains to LB isolates failed to prove the correlation due to incomplete identification of the strains.

We found the frequency of LB is low, albeit with high mortality, and that probiotic strains are an infrequent source of bacteremia. The use of probiotics can be considered generally safe; however, more data that includes hospital use, especially in immunocompromised hosts, in pregnancy, and in special units such as the ICU, are urgently needed.

## Figures and Tables

**Table 1 microorganisms-11-00896-t001:** Risk factors associated with *Lactobacillus* bacteremia.

Risk Factors
Malignancy.
Serious gastrointestinal disorder (i.e., hepatic cirrhosis, cholecystolithiasis, chronic pancreatitis).
Prior hospitalization.
Previous antibiotic treatment.
Surgery.
Preterm neonates with underlying conditions (short bowel syndrome, growth restriction).
Ischemic or ulcerative colitis.
Immunocompromised (HIV, chronic steroids, chemotherapy, or transplant patients).
Use of central venous devices.
Concomitant polymicrobial infections.

**Table 2 microorganisms-11-00896-t002:** Reports of possible *Lactobacillus* spp. bacteremia with no evidence of oral probiotic administration or cases where the blood isolate did not match the probiotic strain.

Identity of Blood Isolate (n)	History of Probiotic Use	Identity of Probiotic Strain	Reference
No history of probiotic use			
*Lcb. rhamnosus* nr (n = 28)	None	Na	[24]
*L. jenseii* nr (n = 1)	None	Na	[26]
*Lpb. plantarum* nr (n = 1)	None	Na	[27]
*Lcb. rhamnosus* GG and *Lcb. rhamnosus* Lcr35 (n = 1)	None	Na	[33]
“*Lactobacillus*” nr (n = 1)	None	Na	[35]
*L. jensenii* nr (n = 1)	None	Na	[45]
*Lcb. casei* nr (n = 1)	None	Na	[46]
*Lpb. plantarum* nr (n = 3)	None	Na	[47]
*Lcb. rhamnosus* non-GG (n = 4)	None	Na	[47]
*Lcb. casei* nr (n = 1)	None	Na	[47]
*L. gasseri* nr (n = 1)	None	Na	[47]
*L. acidophilus* nr (n = 1)	None	Na	[47]
*Lcb. rhamnosus GG ** nr (n = 2)	None	Na	[48]
*Lcb. paracasei* Lp10266 (n = 1)	None	Na	[49]
“*Lactobacillus*” nr (n = 1)	None	Na	[50]
*L. jensenii* nr (n = 1)	None	Na	[51]
*Different strains found*			
*Lcb. casei* non-LBC80R (n = 1)	Yes	*Lcb. casei* LBC80R	[53]
*Lsb. fermentum* nr (n = 1)	Yes	*L. acidophilus* nr	[54]
*L. acidophilus* nr + *Bifido*. *bifidum* nr (n = 1)	Yes	*Lcb. rhamnosus* nr	[55]

* Two cases of roommates with an LB case taking probiotic *Lcb. rhamnosus* GG. Abbreviations: *L.*, *Lactobacillus*; *Lcb.*, *Lacticaseibacillus*; *Lpb.*, *Lactiplantibacillus*; *Lsb.*, *Limosilactobacillus*; Na, not applicable; n, number; nr, not reported.

**Table 3 microorganisms-11-00896-t003:** Cases of Lactobacilli bacteremia confirmed to be associated with the same strain of oral *Lactobacilli* probiotic use.

Probiotic	Method Used	Number of Bacteremia Cases	Lactobacilli Isolate Matched *	Age	Underlying Condition(s)	Recovered after Antibiotic Treatment	Reference	Died
*Lcb. rhamnosus* GG (ATCC53103)	PFGE	2	1	Premature neonates	Short gut syndrome	Yes	No	[42]
*Lcb. rhamnosus* GG (ATCC53103)	16-s rRNA	1	1	11 mon	Short gut syndrome, CVC	Yes	No	[56]
*Lcb. rhamnosus* GG (ATCC53103)	PCR	2	2	6 yr 6 wk	AAD, CVC	Yes	No	[57]
*Lcb. rhamnosus* GG (ATCC53103)	16-s rRNA	1	1	17 yr	Ulcerative colitis	Yes	No	[58]
*Lcb. rhamnosus* GG (ATCC53103)	Rep PCR	1	1	6 day	IUGR	Yes	No	[59]
*Lcb. rhamnosus* GG (ATCC53103)	PFGE	1	1	64 yr	Ulcerative colitis, CAD	Yes	No	[60]
*Lcb. rhamnosus* GG (ATCC53103)	PFGE	2	2	18 days, 3 mon	Premature infants, CAD, CVC	Yes	No	[11]
*Lcb. rhamnosus* GG (ATCC53103)	PCR	1	1	54 yr	Leukemia	Yes	No	[61]
*Lcb. rhamnosus* GG (ATCC53103)	WGS	6	6	1–19 yr	ICU patients	Yes	No	[47]
*Lcb. rhamnosus* GG (ATCC53103)	PCR	1	3	Neonates	ICU, CVC	Yes	No	[48]
*Lcb. rhamnosus* GG (ATCC53103)	GS	1	1	Neonate	ICU, preterm, CVC	Yes	No	[62]
*Lcb. paracasei in a 7-strain blend*	GS	6	1	10 mon–16 yr	Hematopoietic cell transplant, cancer	Yes	No	[38]
*Lpb. plantarum in a 7-strain blend*			2			Yes	No	

* Same strain found using genomic analysis in the blood sample and probiotic. Abbreviations: 16-s rRNA, 16s rRNA gene sequencing; AAD, antibiotic-associated diarrhea; CAD, cardioarterial disease; CVC, central venous catheter; GS, genomic sequencing, method not reported; ICU, intensive care unit; IUGR, intrauterine growth restriction; *L.*, *Lactobacillus*; *Lcb.*, *Lacticaseibacillus*; *Lpb.*, *Lactiplantibacillus*; mon, months old; nr, not reported; PCR, PCR DNA fingerprint assay; PFGE, pulse field electrophoresis; rep-PCR, DNA fingerprint analysis; WGS, whole genome sequencing; wk, weeks; yr, years old.

## Data Availability

Not available.

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
