# Peer review of "Lactobacillus Bacteremia and Probiotics: A Review"

_microorganisms, 2023, doi:10.3390/microorganisms11040896_

Round 1

Reviewer 1 Report

Manuscript: microorganisms-2235238 Lactobacillus bacteremia and probiotics: A review.

The authors reviewed the literature for articles (1980–2023) on the pathogenicity of Lactobacillus spp. bacteremia and reports of probiotics in these patients. The authors updated the present knowledge on epidemiology of Lactobacillus spp. bacteremia and determined the role of probiotics in Lactobacillus bacteremia. Lactobacillus bacteremia is infrequent but has a higher risk of mortality and risk factors include severe underlying diseases, immune system suppression, admission to intensive care units and use of central venous catheters. A variety of Lactobacillus species may cause bacteremia and may or may not be associated with probiotic exposure. The prevalence of Lactobacillus bacteremia is infrequent, but is more common in patients taking probiotics compared to those not taking probiotics. Three probiotics (L rhamnosus GG, L. plantarum, L. paracasei) were directly linked with blood isolates from bacteremia patients using molecular identification assays.

The data analysis methods are correct.

The English of the text is well written and well readable but needs additional checking with a professional translator.

The uniqueness of the text is more than 90% by AntiPlagiarism.NET.

The text contains some misspellings and typos. Also need to expand the part of the discussion.

There are some comments and questions:

1) What is novelty of this manuscript. Everything is known now? 

2) The authors should add to the Conclusions their recommendations for use Lactobacilli in medicine. Maybe it is need to recommend the using postbiotics, metabiotics and pharmabiotics for human treatment. The authors should say what choice to make for most people to minimize the risks as much as possible - take or refuse probiotics.

3) This manuscript I think should describe not only harm but also the benefit of lactobacilli. Add one chapter about benefits of lactobacilli. 

4) Add chapter about postbiotics, metabiotics and pharmabiotics - as a way to safely treat a human.

5) The Discussion part is very small. Add discussion and hypothesis on reason and possible molecular mechanisms of Lactobacillus bacteremia.

6) Lines 34-35 and 40-42 - after the sentences - They are widely distributed in nature and have multiple commercial uses, especially in the fermentation of cheese and other dairy products [2] - and - Probiotics are commonly used for the prevention of Clostridioides (Clostridium) difficile infections or the prevention of antibiotic-associated diarrhea among other diseases [1,10,11]. - add additional citation (Danilenko et al., 2021)

7) Add to the References: Danilenko, V.N.; Devyatkin, A.V.; Marsova, M.V.; Shibilova, M.U.; Ilyasov, R.A.; Shmyrev, V.I. Common inflammatory mechanisms in COVID-19 and Parkinson’s diseases: the role of microbiome, pharmabiotics and postbiotics in their prevention. J Inflamm Res 2021, 14, 6349–6381, doi:10.2147/JIR.S333887.

8) Line 308 - electrophoriss - should be electrophoresis.

Please improve the manuscript according to the above comments.

Author Response

1) What is novelty of this manuscript. Everything is known now? 

In the past publications on the risks of probiotics, the case reports were often not substantiated by sensitive and precise identification techniques for Lactobacilli strains. In our systematic review, we gathered case reports and case series and we report publications that did or did not use the more recent specific identification techniques based on genomic assays (microbiologic culture techniques are not precise enough) to determine if the strains of the blood isolates and the oral probiotic strain being taken were identical. This has not been methodically done before. Also, in previous articles, oral probiotics were blamed for the Lactobacilli bacteremia, but the patient had no history of probiotic use. This careful review allows readers to determine which probiotic strains actually might have been the source.

2) The authors should add to the Conclusions their recommendations for use Lactobacilli in medicine. Maybe it is need to recommend the using postbiotics, metabiotics and pharmabiotics for human treatment. The authors should say what choice to make for most people to minimize the risks as much as possible - take or refuse probiotics.

Please note that “Postbiotics are defined (also known as metabiotics, biogenics, or simply metabolites) as soluble factors (metabolic products or byproducts), secreted by live bacteria, or released after bacterial lysis providing physiological benefits to the host. We focused here only on oral Lactobacilli probiotic cases. Oral living microbes may cause infection, but dead or inactivated probiotics or prebiotics, or postbiotic, metabiotics, etc. may leak from the gut but are incapable of causing an active infection. Therefore we focused only on living microbial probiotics.

3) This manuscript I think should describe not only harm but also the benefit of lactobacilli. Add one chapter about benefits of lactobacilli.

We have added benefits in the introduction. “Many lactobacilli are commensal flora found within the human gastrointestinal tract and the female genitourinary tract, being beneficial in helping protect from chronic diseases such as inflammatory bowel disease and being an important member of the gut and vaginal microbiome. Further, since most Lactobacillus species are probiotic microorganisms, they produce enzymes that display antibiotic, anticancer, and immunosuppressant properties.”

4) Add chapter about postbiotics, metabiotics and pharmabiotics - as a way to safely treat a human.

Please note that “Postbiotics are defined (also known as metabiotics, biogenics, or simply metabolites) as soluble factors (metabolic products or byproducts), secreted by live bacteria, or released after bacterial lysis providing physiological benefits to the host. We focused here only on oral Lactobacilli probiotic cases. Oral living microbes may cause infection, but dead or inactivated probiotics or prebiotics, or postbiotic, metabiotics, etc. may leak from the gut but are incapable of causing an active infection. Therefore we focused only on living microbial probiotics.

5) The Discussion part is very small. Add discussion and hypothesis on reason and possible molecular mechanisms of Lactobacillus bacteremia.

The possible mechanisms of LB should be addressed. As a risk factor of LB is presence of a central venous catheter, direct contamination of the central line with Lactobacillus spp. or with stool containing the strain could lead to LB. Further, most patients that have LB have severe underlying diseases that predispose them to bacteremic complications, including LB. It has been noted that adhesion to intestinal surfaces is one of the main selection criteria for probiotic bacteria as well as lack of platelet aggregation, as this has been shown to be harmful for the host, potentially leading to endocarditis. Adhesion allows preservation in the gut and increases the opportunity for probiotic species to influence the host’s microbial balance and gastrointestinal immune system.

6) Lines 34-35 and 40-42 - after the sentences - They are widely distributed in nature and have multiple commercial uses, especially in the fermentation of cheese and other dairy products [2] - and - Probiotics are commonly used for the prevention of Clostridioides (Clostridium) difficile infections or the prevention of antibiotic-associated diarrhea among other diseases [1,10,11]. - add additional citation (Danilenko et al., 2021)

We have added this reference.

7) Add to the References: Danilenko, V.N.; Devyatkin, A.V.; Marsova, M.V.; Shibilova, M.U.; Ilyasov, R.A.; Shmyrev, V.I. Common inflammatory mechanisms in COVID-19 and Parkinson’s diseases: the role of microbiome, pharmabiotics and postbiotics in their prevention. J Inflamm Res 2021, 14, 6349–6381, doi:10.2147/JIR.S333887.

We have added this reference.

8) Line 308 - electrophoriss - should be electrophoresis.  

We have changed this.

Reviewer 2 Report

In this article, the study authors review the literature related to bacteremia due to Lactobacillus genus species and the role that probiotics may play in it. Overall, it is a commendable and comprehensive review of the literature of this rare occurrence. I just have a few comments as below, to help further increase the academic value of this article.

Major Comments

1. The authors summarize some of the risk factors that are associated with Lactobacillus bacteremia (Page 3 of 17). For increased readability, it would be helpful if a brief table highlighting these risk factors was provided.

2. The study authors note that there is a risk of Lactobacillus bacteremia for patients and their roommates with central venous catheters during opening of capsules and reconstituting the powder to liquid for tube feedings (Page 12 of 17, lines 388-391). This is an interesting point that deserves further discussion from an Infection Control standpoint. Should healthcare facilities discourage this? Have studies looked at this potential nosocomial route of spread? These are important points to bring up in the discussion.

Minor Comments

1. Page 3 of 17, line 94 - the authors have reported that in the study by Salminen et al, “severe underlying diseases” was a significant predictor of mortality. It would be educative if they elaborated further on the comorbidities that were studied.

2. Page 3 of 17, line 131 – there is a minor formatting error in the start of the paragraph

3. Page 8 of 17, Table 2 – for the third entry in the table, the study authors have written one comorbidity as AAD. Did they mean CAD? Please clarify this

4. Page 13 of 17, line 423 – consider changing “compromised host” to “immunocompromised host”

Author Response

  1. The authors summarize some of the risk factors that are associated with Lactobacillus bacteremia (Page 3 of 17). For increased readability, it would be helpful if a brief table highlighting these risk factors was provided.

We have added a table (Table 1).

  1. The study authors note that there is a risk of Lactobacillus bacteremia for patients and their roommates with central venous catheters during opening of capsules and reconstituting the powder to liquid for tube feedings (Page 12 of 17, lines 388-391). This is an interesting point that deserves further discussion from an Infection Control standpoint. Should healthcare facilities discourage this? Have studies looked at this potential nosocomial route of spread? These are important points to bring up in the discussion.

There is not much published on this, but we have added this in the discussion: “. We recommend taking a precautionary approach to using probiotics in the ICU in patients with central venous catheters. The risk may be because powder more easily distributes to contaminate central venous catheters. The risk of contamination is likely both from the air and contaminated hands, which can cause a translocation to a central line catheter where the organisms have direct entry into the systemic circulation.”

Minor Comments

  1. Page 3 of 17, line 94 - the authors have reported that in the study by Salminen et al, “severe underlying diseases” was a significant predictor of mortality. It would be educative if they elaborated further on the comorbidities that were studied.

We have included this: “Salminen et al. reviewed the risk factors and outcomes for LB using multivariate analysis and found severe underlying diseases, which were primarily malignancies or serious gastrointestinal disorders…”

  1. Page 3 of 17, line 131 – there is a minor formatting error in the start of the paragraph.

We have taken care of this.

  1. Page 8 of 17, Table 2 – for the third entry in the table, the study authors have written one comorbidity as AAD. Did they mean CAD? Please clarify this.

We have added the footnote definition of AAD (antibiotic-associated diarrhea) in Table 2.

  1. Page 13 of 17, line 423 – consider changing “compromised host” to “immunocompromised host”

We have taken care of this.